# Possible Association between Selected Tick-Borne Pathogen Prevalence and *Rhipicephalus sanguineus* sensu lato Infestation in Dogs from Juarez City (Chihuahua), Northwest Mexico–US Border

**DOI:** 10.3390/pathogens11050552

**Published:** 2022-05-07

**Authors:** Diana M. Beristain-Ruiz, Javier A. Garza-Hernández, Julio V. Figueroa-Millán, José J. Lira-Amaya, Andrés Quezada-Casasola, Susana Ordoñez-López, Stephanie Viridiana Laredo-Tiscareño, Beatriz Alvarado-Robles, Oliver R. Castillo-Luna, Adriana Floriano-López, Luis M. Hernández-Triana, Francisco Martínez-Ibáñez, Ramón Rivera-Barreno, Carlos A. Rodríguez-Alarcón

**Affiliations:** 1Departamento de Ciencias Veterinarias, Universidad Autónoma de Ciudad Juárez, Anillo Envolvente y Estocolmo s/n Colonia Progresista AP 1729-D Cd. Juárez, Chihuahua 32310, Mexico; diana.beristain@uacj.mx (D.M.B.-R.); aquezada@uacj.mx (A.Q.-C.); ordonez.susana@yahoo.com.mx (S.O.-L.); balvarad@uacj.mx (B.A.-R.); al122014@alumnos.uacj.mx (O.R.C.-L.); al136851@alumnos.uacj.mx (A.F.-L.); rrivera@uacj.mx (R.R.-B.); 2Departamento de Ciencias Químico-Biológicas, Universidad Autónoma de Ciudad Juárez, Anillo Envolvente y Estocolmo s/n Colonia Progresista AP 1729-D Cd. Juárez, Chihuahua 32310, Mexico; javier.garza@uacj.mx (J.A.G.-H.); stephanie.laredo@uacj.mx (S.V.L.-T.); 3CENID-Salud Animal e Inocuidad, Instituto Nacional de Investigaciones Forestales, Agrícolas y Pecuarias, Cuernavaca-Cuautla 8534, Progreso, Jiutepec 62574, Mexico; figueroa.julio@inifap.gob.mx (J.V.F.-M.); lira.juan@inifap.gob.mx (J.J.L.-A.); 4Virology Department, Vector Borne Diseases Research Group, Animal and Plant Health Agency (APHA), Woodham Lane, New Haw, Addlestone, Surrey KT15 3NB, UK; luis.hernandez-triana@apha.gov.uk; 5Departamento de Ectoparásitos y Dípteros, CENAPA-SENASICA, Jiutepec 62550, Mexico; francisco.martinez@senasica.gob.mx

**Keywords:** Mexico–US border, *Otobius megnini*, *Rhipicephalus* *sanguineus*, tick-borne pathogens, ticks

## Abstract

Tick-borne bacterial pathogens (TBBPs) show a worldwide distribution and represent a great impact on public health. The brown dog tick (*Rhipicephalus sanguineus*) is a vector of several pathogens that affect dogs and sometimes humans as well. In addition, TBBPs represent a diagnostic challenge and imply financial resources and medical treatment for long periods of time. In the present study, *R. sanguineus* s. l. was identified as the main tick species naturally parasitizing dogs that inhabit. Juárez City, Chihuahua, in the Paso del Norte region, Mexico–US Border, representing 99.8% of the cases. Additionally, an end-point PCR was performed to search for whether pathogens in *R. sanguineus* s. l. can transmit in DNA extracted from ticks and dog blood samples. This is the first molecular detection of *Rickettsia rickettsi* infecting domestic dogs in Mexico; however, other pathogens were also identified, such as *Ehrlichia canis* and *Anaplasma platys* in both ticks and dog blood samples, while *Anaplasma phagocytophilum* was identified only in dog blood samples. Moreover, co-detection in tick pools and co-infection in the analyzed dog blood samples could be found. Similarly, this research showed that dogs were found mostly parasitized by adult female ticks, increasing the possibility of transmission of *E. canis*.

## 1. Introduction

Tick-borne bacterial pathogens (TBBPs) such as ehrlichiosis, borreliosis, anaplasmosis, and rickettsiosis are well-known diseases of veterinary and medical importance [1]. The etiological agents of TBBPs are zoonotic bacteria that significantly impact public health, such as *Ehrlichia canis*, *Borrelia* spp., *Rickettsia rickettsii,* and *Anaplasma* spp., among others [2,3,4]. Ixodidae (hard ticks) and Argasidae (soft ticks) families (Acari order) can transmit TBBPs [5,6]. Both groups of ticks are obligate ectoparasites of vertebrates, including birds, mammals, reptiles, and amphibians, with humans as incidental hosts [7,8]. Several epidemiological studies using domestic animals as sentinels have been utilized to monitor infectious diseases, including TBBPs [9]. Those epidemiological methods can provide substantial surveillance information to identify dynamics in the infection and health status of the animal and human populations, determine patterns in a pathogen mix [10], provide evidence of disease absence, or estimate the prevalence of a given pathogen [11]. Moreover, they are essential to controlling and preventing diseases timely in a relatively inexpensive manner. Thus, dogs can play a valuable role as sentinel hosts for monitoring diseases [12] since they live in close contact with humans and livestock, and are susceptible to many emerging or re-emerging human TBBPs [13]. Therefore, they represent an essential human epidemiological surveillance tool to explore the variability in the TBBPs [9].Conversely, ticks can be used to monitor TBBPs. The use of hematophagous arthropods to survey vertebrates for the presence of infectious disease agents is called xenosurveillance and is well documented [14,15]. Hence, knowledge of the infestation with ticks can be useful to estimate associations between tick density and the prevalence of TBBPs, and to understand the ecology of pathogen transmission [16]. 

Currently, there is no epidemiological information about the infestation with ticks that naturally parasitize dogs and the prevalence of TBBPs in Juarez City, in the Paso del Norte border region. Therefore, to expand our knowledge about the epidemiology of ticks in this binational region, this study has the following objectives: (1) to estimate tick infestation load and species of ticks in both free-ranging dogs (FRD) and home dogs and (2) to identify the association among the prevalence of TBBPs with tick infestation and other risk factors of dogs in Juarez City, Chihuahua, on the Mexico–U.S. border. 

## 2. Results

### 2.1. Morphological and Molecular Identification of Ticks

Out of the total ticks collected in this study, 99.82% (*n* = 1688) were morphologically identified as *Rhipicephalus sanguineus* s.l. (Figure 1, Figure 2, Figure 3 and Figure 4), whereas the remaining 0.18% (*n* = 3) were identified as *Otobius megnini* (Figure 5). Morphological identification was corroborated using DNA barcode analyses. The DNA barcodes (not shown) corresponded to those reported in genomic databases [17,18].

### 2.2. Tick Infestation between 2013 and 2014

In the first 2-year period of the study (2013–2014), 979 ticks corresponding to *R. sanguineus* s. l., were collected (548 and 431 ticks in 2013 and 2014, respectively) from 237 dogs inspected (137 FRD and 100 home dogs: designated as two categories). Table 1 show the infestation load of each life stage of *R. sanguineous* s. l. by year both of the FRD and home dogs. The LSM of total ticks infesting FRD was significantly higher than that of home dogs. A similar situation was seen in females, males, blood-fed females, and unfed females. However, in nymphs and larvae, no statistical differences were present. Table 2 show the Linear Square mean (LSM) ± Standard error (SE) for each life metamorphic stage of ticks per year and both dog categories (FRD/home dogs), including the *p*-values from T-student and goodness of fit of chi-square/df.

### 2.3. Prevalence of Tick-Borne Pathogens in Dogs

In the 2015 study, infection with *E. canis* was detected in 53.60% (104/194), *A. platys* in 24.74% (48/194), *A. phagocytophilum* in 12.88% (25/194), and *R. rickettsii* in 5.67% (11/194) of dogs. Co-infections of two to three pathogens were detected in 21.64% (42/194) of dogs, where *E. canis* and *A. platys* were 9.27% (18/194), *A. platys* and *A. phagocytophilum* were 3.6% (7/194), *E. canis* and *R. rickettsii* were 1.54% (3/194), *E. canis* and *A. phagocytophilum* were 1.03% (2/194), *E. canis*, *A. platys*, and *A. phagocytophilum* were 4.64% (9/194), *E. canis*, *A. phagocytophilum*, and *R. rickettsii* were 1.03% (2/194), and for *E. canis*, *A. platys,* and *R. rickettsii* were 0.52% (1/194). Finally, *B. burgdorferi* s. l. was not detected by PCR in dogs.

### 2.4. Prevalence of Tick-Borne Pathogens in Ticks

In the case of TBBPs detection in the tick pools, 72% (43/60) were positive for some of the pathogens analyzed, and 28% (17/60) were negative by PCR molecular detection. The highest prevalence estimated was for *E. canis*, with 66.6% (40/60) of the positive samples, followed by *A. platys* with 8.3% (5/60) and *R. rickettsii* with 5% (3/60). All tick pools were PCR negative for *A. phagocytophilum* and *B. burgdorferi* s. l. Pathogen co-detection was also observed in tick pools, with the most common ones being *E. canis* and *A. platys* with 6.6% (4/60) and *E. canis* with *R. rickettsii* in 1.6% of the pools (1/60).

Access numbers of sequences are available at NCBI as follows: *E. canis* MK386937, MK386938, and MK386938.1; *A. platys* MK386768; and finally *R. rickettsii* MK350322 and MK350322.1. *A. phagocytophilum* could not be sequenced.

### 2.5. Association of the Prevalence TBBPs among Dog Risk Factors

Since *E. canis* infection showed more than 50% prevalence in dogs (*n* = 104), only this pathogen was used to understand the association between TBBPs and risk factors (breed, tick infestation sites on dogs, age, sex, total ticks, females, males, blood-fed females, unfed females, nymphs, and larvae) (Appendix A). The results of GLIMMIX for multivariate logistic regression showed a good fit for the model (Pearson’s Chi-square/degrees of freedom = 149.47/138 = 0.96). The unique predictor variable that showed a trend of *E. canis* prevalence was represented by female ticks (α = 0.09; using a critical level of α = 0.10 as selection criteria) (Figure 6); the remaining explanatory variables were associated with alpha levels higher than 0.10. This relationship was detected when the number of total ticks varied significantly as a response variable on male ticks per dog. Therefore, the association indicates that for each female tick, the probability of *E. canis* increased (Figure 7).

## 3. Discussion

In previous studies in Mexico, the seroprevalence of TBBPs detected in both ticks and animals was common [19,20,21,22,23], and surveys on the diversity of ticks were carried out [21,24]. However, this research represents the first systematic study performed at the Mexico–US border, where the co-infestation with *R. sanguineus* s. l. and *O. megnini*, naturally parasitizing domestic dogs and FRD, the prevalence of TBBPs, and the association among the infestation with *R. sanguineus* s. l. males and *E. canis* infection in dogs were identified. Likewise, this is the first report in this region of active infection for *R. rickettsii* in dogs and co-infection of other TBBPs. It is essential to highlight that this study, like others of global relevance, reflects the importance of carrying out epidemiological studies in dogs due to the high probability of *R. sanguineus* s. l. representing a vector to transmit TBBPs to humans [3,12].

In the present study, we observed two species of ticks naturally parasitizing dogs. According to taxonomic characteristics, these species were identified as *R. sanguineus* s. l. and *O. megnini* (Figure 1, Figure 2, Figure 3, Figure 4 and Figure 5). The DNA barcodes corresponded to those reported in genomic databases [17,18].

Different studies were performed in Mexico to identify ticks that parasitize dogs. For example, Cruz-Vázquez et al. (1998) [25] and Tinoco-Gracia et al. (2009) [26] only reported *R. sanguineus*, while it was the most predominant species in studies conducted by Galaviz-Silva et al. (2013) [27] and Ojeda-Chi et al. (2019) [28].

Although *R. sanguineus* is native to Africa, it has a cosmopolitan distribution due to the migratory movements of humans and their dogs. Global studies showed that *R. sanguineus* is a species well adapted to urban areas, where it is the most common [3,29,30]. This is in accordance with our results, i.e., the high prevalence of this tick is in the urban area of Juarez. In addition, based on these studies, we confirm that *R. sanguineus* s. l. is well adapted to the arid region of northern Mexico, better than other species of ticks [31,32,33].

According to the results shown in Table 2, there were no significant differences between sexes, developmental stages, or feeding degrees of *R. sanguineus* s. l. ticks, most likely because this study was performed in the seasons with a higher incidence of ticks and the diseases they transmit [3,34]. Regarding the finding of *O. megnini* parasitizing one of the dogs included, a similar case was already described in a dog from the north-central zone of Mexico [35].

In this research, the TBBP with the highest prevalence was *E. canis*, with 53.6% of positive dogs. This prevalence is superior in comparison with other studies using the PCR assay, even in Caribbean countries with a tropical climate, such as Grenada, with 24.7% of positive dogs to *E. canis* that were 100% parasitized by *R. sanguineus* [36]. Furthermore, the prevalence is also high when compared with other studies in Mexico; for example, Almazan et al. (2016) reported a prevalence of 10% [33]. In another study, where the prevalence was determined twice, there was a prevalence of 29.26% (72/246) for *E. canis* at the first sampling event, whereas at the second sampling event, it was 38.46% (20/52) [37]. Finally, Pat-Nah et al. (2015) determined a 36% prevalence of this disease in Yucatan, Mexico [38]. Similarly, in the neighboring state of Texas, USA, the prevalence of this disease was 1.62% (19/1171) [39], indicating that the prevalence of *E. canis* in our study is the highest reported so far.

The second-prevalent pathogen of TBBPs was *A. platys*, the etiologic agent of canine infectious cyclic thrombocytopenia, with 24.74% of cases (48/194). Studies on this pathogen in this region are also uncommon: the first molecular report of *A. plat*ys in Mexico was published in 2013 [40] and confirmed in 2015 [33]; likewise, a study performed in Texas, a state bordering Juarez, Mexico, found a prevalence of this pathogen in 0.17% of dogs (2/1171) [39] and in areas relatively close to the region of this study. It was also reported in white-tailed deer (0.4%) [41]. However, this species was described in several countries of the American continent, along with other tick-borne pathogens [42,43,44].

The etiologic agent of human granulocytic anaplasmosis, *A. phagocytophilum*, was detected in 12.88% of the dogs (25/194). In a study performed in southern Mexico, a prevalence of 27% was found [45], which is higher than that of the present study. In addition, other studies conducted in the same area as the present study and nearby places showed varying results. For example, Modarelli et al. (2019) did not detect this pathogen in dogs [39]. Furthermore, *A. phagocytophilum* was reported in horses (0.8% prevalence) from the same geographic area [46] and in ticks from dogs in Chihuahua state [47] which is the same state that the area where the present study was performed. These differences could be attributed to the conditions in which each investigation was carried out. Each analysis varied in sample size, sampling time, and species included, which may further support the observed disparities.

The lowest prevalence of tick-borne pathogens found in the present research was observed for *R. rickettsii* (5.67%) (11/194), the etiologic agent of Rocky Mountain Spotted Fever (RMSF). Although it was present in a low proportion, the presence of this pathogen is critical since rickettsiosis is a zoonotic disease and has caused death in humans in North America [48,49,50]. In fact, the state of Chihuahua is now the first national place in the presentation of RMSF in humans in Mexico [51]. Additionally, RMSF is considered the most important rickettsiosis in America, with high mortality [52]. In Mexico, cases of human rickettsiosis have increased in recent years and include affections in children. These data are also applicable to the state of Chihuahua, where this disease had a mortality of 35.92% (334/120) during the 2013–2019 period [51].

In dogs, RMSF was detected by the Indirect Fluorescent Antibody test in Arizona (prevalence of 53.4 to 66.4%) [53] and Oklahoma (54.6% to 88.5%) [54]. However, in these investigations, the pathogen was not detected by PCR. On the other hand, PCR studies were carried out on dogs in Texas, but this pathogen was not identified in blood samples [39]. In another study carried out in Carolina, *R. rickettsii* was only detected by PCR in 1.01% of the dogs (1/99), whereas via serology, it was detected in 21.21% (21/99) [55]. In Yucatán, Mexico, Ojeda-Chi et al. (2019) carried out an investigation that included blood samples from 246 dogs, in which they could not find the DNA of this rickettsia [28]. Nevertheless, in Mexico, the presence of *R.*
*rickettsii* was identified by PCR in humans [56] and ticks [22,57,58]. This is the first study performed in Mexico where *R. rickettsii* is identified in the blood of dogs by PCR, indicating an active infection.

Finally, it was impossible to detect *B. burgdorferil* s. l., the etiological agent of Lyme disease, which was neither found in dogs nor ticks analyzed in this study. This finding could perhaps be attributed to the particular characteristics of *B. burgdorferi*, which is highly transitory in blood. Lyme disease is one of the most important zoonoses in the US, with a presentation of around 300,000 cases per year in humans [59], and was also described in dogs and ticks at the US–Mexico border [27,60,61,62]. The absence of this pathogen in our study might be explained by the fact that the primary vector of *Borrelia burgdorferi* is another tick, *Ixodes* spp. not *R. sanguineus* s. l. or *O. megnini*, which are the ticks that were identified in this area.

The co-detection of tick-borne diseases are frequent in small animal practice and are described in both animals and ticks via PCR [19,22,33,39,53]. In some cases, they are detected in dogs using other diagnostic techniques, such as ELISA [63]. Similarly, in other domestic species in the Juárez region (Mexico–US Border), it was possible to detect co-infection in horses and ticks by PCR [46]. Additionally, co-infection of tick-borne pathogens was also observed in wild species in nearby areas [41,45]. The co-infection with the highest prevalence in the analyzed dogs was that of *E. canis* and *A. platys* (9.27%; 18/194), most likely because both pathogens share the tick vector and their primary host is the dog; this co-infection was even reported in areas close to where the present study was conducted [39]. The other observed co-infections in dogs were present at a lower percentage, and in most of these cases, *E. canis* was involved (18.04%; 34/194). This could be because more than 99% of the identified ticks in this study were *R. sanguineus* s. l., which has the dog as its primary host and is the main vector of *E. canis*, although it is also associated with the presence of different rickettsia. Additionally, only dogs with tick infestation at the time of inspection were included in this study, increasing the possibility of finding active infections.

The detection of *E. canis* in ticks in this study was only performed in pools of adult ticks. However, we noted that for each female found in dogs, the possibility of transmission of *E. canis* increased. These results differ from those described by other researchers who observed that high infestation with males ticks could influence the prevalence of *E. canis* because of the transstadial transmission of *E. canis* to dogs by ticks that feed on them during the immature stages; the male *R. sanguineus* may be able to transmit *E. canis* in the absence of female ticks [63]. During the different stages of its life cycle, this tick ingests blood; larvae and nymphs of *R. sanguineus* do it to molt. In contrast, adults feed on blood for reproduction purposes. On the other hand, females need large amounts of blood to produce eggs, increasing their body weight up to 100 times after feeding [64]. In contrast to what happens with male ticks, adult males only consume blood for spermatogenesis and to complete the reproductive process [65]. The fact that the present study showed an increase in the possibility of transmission of *E. canis* in dogs parasitized by female ticks might be due to the blood-feeding period of females being longer than males. However, more research on this point is needed.

Additionally, in a study by Ipek et al. (2018), *E. canis* was detected by PCR in 3 out of 12 (25%) male pools and 6 of 14 (42.8%) female pools obtained from dogs infected with *E. canis* [66]. Similarly, in another investigation in Colombia, the pools of male *R. sanguineus* presented a lower percentage of positives to *E. canis* than the pools of female ticks: 11.8% in males versus 16.3% in female ticks [67].

## 4. Materials and Methods

### 4.1. Ethical Approval and Informed Consent

This study was reviewed and approved by the Ethical and Bioethical Committee of the Autonomous University of Juarez (CIBE-2016-1-16), Mexico, and was performed in compliance with Mexican and American guidelines for research animals (Guide for the Care and Use of Laboratory Animals in National Resource Council, 2011). All owners of the sampled dogs signed an informed consent form as authorization for the collection of the ticks and blood samples. Molecular analysis was carried out at the Laboratory of Veterinary Clinical Pathology and Molecular Biology of the Autonomous University of Juárez and the Babesia Laboratory Unit at the National Center for Disciplinary Research in Animal Health and Safety (CENID-SAI) of the National Institute of Forestry, Agricultural, and Livestock Research (INIFAP), Mexico.

### 4.2. Study Area and Samples

This study was conducted in Juarez, in the Northwest of Mexico (31°43′59″ N; 106°28′59″; at 1120 m above sea level) in the Chihuahua Desert [68]. Juárez City has about one and a half million inhabitants (INEGI, 2020). The climate is very dry, extreme, and semi-desertic, with highs of 40 °C during summer and −19 °C in winter. Juarez is a binational metropolitan area connected with El Paso County, Texas, and Donna Ana county, New Mexico, US. These three cities work together to promote commerce, tourism, and industry throughout the region by signing a sister cities agreement, where the migration of people accompanied by domestic animals (pets) is frequent [69].

#### 4.2.1. Dog Recruitment and Sampling

The only inclusion criterion for the study was that dogs presented ticks at the time of inspection. This is in order to identify the tick species present in the locality. Between January 2013 and December 2014, 137 free-roaming dogs (FRD) were recruited from the municipal anti-rabies center (MARC). Additionally, 100 randomized home dogs were selected from various veterinary clinics (VC). In both cases, when dogs arrived at MARC or VC, a general inspection was performed, and it was at that moment that the ticks were captured. It was not considered whether dogs showed any signs of TBBPs-associated disease or any other, only the presence of ticks. In 2015, 195 home dogs were recruited only from VC. Figure 8 show the location of the MARC and the distribution of the neighborhoods, including the VCs, in Juarez. A questionnaire was filled in by all dog holders only in 2015 to record information about the household address, breed, age, sex, and clinical history. Blood samples (only for dogs in 2015) were collected from cephalic or jugular venopunction into BD Vacutainer^®^ EDTA K2 tubes; the cell package was separated prior to freezing at −20 °C for further analyses. To detect the TBBPs in ticks and correlate these results with the bacteria found in dogs, ticks from each dog were collected using cotton impregnated with ethanol and entomological forceps. Ticks were stored in plastic bottles with 90% ethanol and frozen (−20 °C) until their identification and/or analysis at the laboratory.

#### 4.2.2. Association among the Prevalence of TBBPs and Tick Stage Risk Factors

To estimate which risk factors influenced the probability of TBBD prevalence (Y = 1, infected dogs; Y = 0, uninfected dogs), GLIMMIX with Multiple Logistic Regression was constructed. Breed, tick infestation sites on dogs (head, ears, neck, body trunk, chest, abdomen, armpits, and legs), age (puppy, adult, and geriatric), sex (male and female), total ticks, females, males, blood-fed females, unfed females, nymphs, and larvae were included as predictors.

#### 4.2.3. Tick Sampling and Identification

From March 2014 to February 2015, ticks were removed from dogs and morphologically identified and grouped according to their metamorphic stages (females, males, blood-feed females, unfed females, nymphs, and larvae), preserved in 70% ethanol, and kept at −20 °C until identification. The identification of tick species was performed using the taxonomic keys established by Martínez-Ibañez (2015) [70]. The DNA barcoding protocols were used to support morphological identification; a 98% sequence similarity between our specimens and those of GenBank and BOLD was considered for correct identification [71,72,73].

The tick counts from both years were found to conform to a negative binomial distribution and were thus examined using a GLIMMIX model. Here, separate sets of GLIMMIXs were run according to each metamorphic stage. The general model in each analysis was as follows: **ticks = year + treat + error** (where ticks = metamorphic stages, year = 2013/2014; treat = FRD/home), which fit the data well with no evidence of over-dispersion [74].

### 4.3. DNA Extraction from Tick Pools and Blood Dog Samples

The DNA extraction from dog blood samples was performed individually, while DNA extraction from ticks was performed by making 60 pools of 5 ticks each taken randomly from dogs with a positive result to any of the pathogens studied. To form the pools, female and male ticks, both adults, were included, as it was the most observed stage. Tick pools were homogenized to perform DNA extraction. In both cases, tick pools and dog blood samples, the extraction was performed using the UltraClean BloodSpin DNA Isolation Kit^®^ (MoBio Laboratories Inc., Carlsbad, CA, USA; Cat. No. 12200-S) following the manufacturer’s protocol.

### 4.4. PCR Amplification and Sequencing

To determine infection by *Ehrlichia canis*, *Anaplasma phagocytophilum, Anaplasma platys*, *Rickettsia rickettsii*, and *Borrelia burgdorferi s.l.* in tick pools and dog blood samples, species-specific PCRs were performed targeting 16S ribosomal RNA genes and the gene ompA coding the outer membrane protein A of *R. rickettsii*, as described elsewhere [75,76,77,78,79]. Table 3 show the oligonucleotide sequences and the PCR conditions. Each 25 µL PCR reaction contained GoTaq Green Master mix (PROMEGA^®^, No, Cat M7123., Fitchburg, WI, USA) 12.5 µL, DNA extracted from tick pools or dog blood samples 5 µL, forward oligonucleotide (10 pmol) 1 µL, reverse oligonucleotide (10 pmol) 1 µL, and nuclease-free water 5.5 µL. The DNA positive controls of *B. burgdorferi* s. l. and *R. rickettsii* were donated by Dr Luis Tinoco-Gracia. The DNA positive controls of *E. canis*, *A. platys* and *A. phagocytophilum* were provided by the Babesia Laboratory Unit of CENID-SAI of the National Institute of Forest, Agricultural, and Livestock Research, Mexico. As negative controls for PCR reactions, we used nuclease-free water along with uninfected dog genomic DNA.

The PCR was performed in a C1000 Touch™ Thermal Cycler (Biorad Laboratories, Hercules, CA, USA); PCR products were analyzed by agarose gel electrophoresis stained with 1% ethidium bromide and examined by a gel documentation system (BioDoc-it, 220; UVP Imaging System; Upland, CA, USA). The PCR products amplified from dog blood or tick pools were purified using the Wizard^®^ SV Gel and PCR Clean-Up System (No. Cat A9281, Madison WI, USA). To validate the identity of the TBBPs molecularly detected, three randomly selected positive samples from each pathogen were purified and sent to Eurofins Genomics Laboratory (Louisville, KY, USA) for bi-directional sequencing. The obtained sequences were edited using the MEGA software v. 7.0 and subsequently aligned with all other sequences from the GenBank database in nBLAST to obtain the percentage of identity; a sequence identity of 98% was considered for correct identification [80].

### 4.5. Statistical Analysis

All statistical analyses were performed using the Statistical Analysis System (SAS 9.4, 2013) OnDemand for the Academics online version. The total count of each life metamorphic stage of ticks was used to estimate the tick infestation load per dog. The generalized linear mixed model (under the GLIMMIX procedure) was used to fit the infestation with each life metamorphic stage of ticks collected to a negative binomial distribution; the least square mean (LSM) of each life metamorphic stage of ticks was then calculated as a response variable (RV) in a negative binomial regression and subsequently compared between FRD and home dogs [74]. The statistical significance of collection among FRD and home dogs was determined using the Student *t*-test for independent samples.

## 5. Conclusions

In this study, we identified *R. sanguineus* s. l. as the main tick that naturally parasitizes both home dogs and free-ranging dogs in Juarez City at the Mexico–US border. Likewise, it was possible to demonstrate the presence of some tick-borne pathogens such as *E. canis* and *A. platys* DNA in ticks and home dogs and *A. phagocytophilum* only in home dogs. It is also important to highlight the presence of co-infections in just over 20% of the patients studied as well as co-detection in the tick pools analyzed. Moreover, this is the first report in Mexico that demonstrates the presence of *R. rickettsii* in the blood of dogs, highlighting the importance of monitoring these findings as this disease is zoonotic. Additionally, it was possible to establish the association between infestation with *R. sanguineus* s. l. females and *E. canis* infection in dogs.

## Figures and Tables

**Figure 1 pathogens-11-00552-f001:**
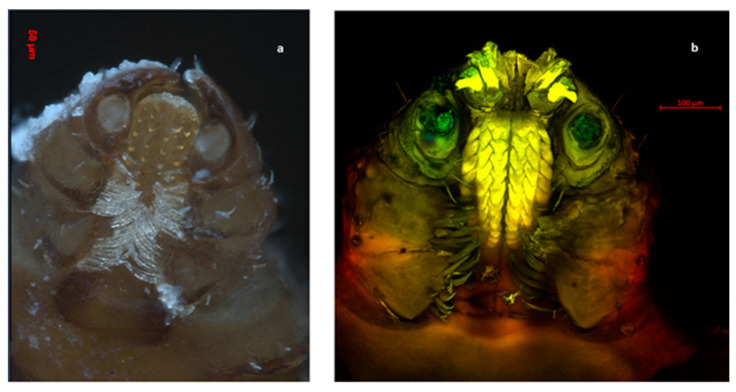
Ventral view of the oral apparatus of an *R. sanguineus* s. l. tick. The hypostome shows four rows of teeth as well as the depression of the four joints present in the pedipalp. (**a**) Photography was carried out with an Axio Zoom V6 microscope. (**b**) Photography was carried outobtained with an LSM 700 confocal scanning microscope.

**Figure 2 pathogens-11-00552-f002:**
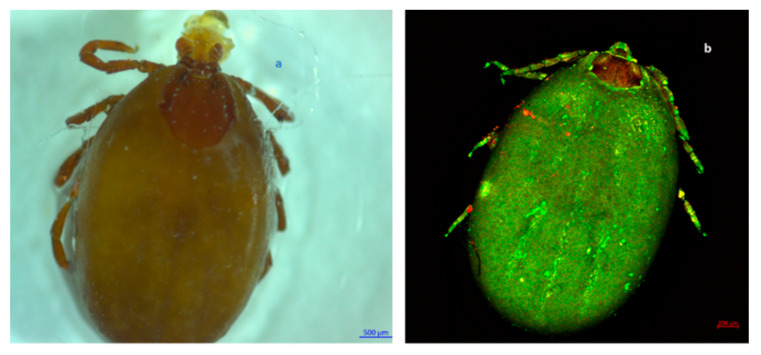
Dorsal view of a fully blood-engorged female of *R. sanguineus* s. l. female. Observe the basis of the hexagonal-shaped gnathostome and the scutum in the first third of the body oridiosoma. In addition, ocelli are present on the sides of the coat of arms (rudimentary eyes). (**a**) Photography obtained with a Stemi 200C microscope. (**b**) Photography obtained with an LSM 700 confocal scanning microscope.

**Figure 3 pathogens-11-00552-f003:**
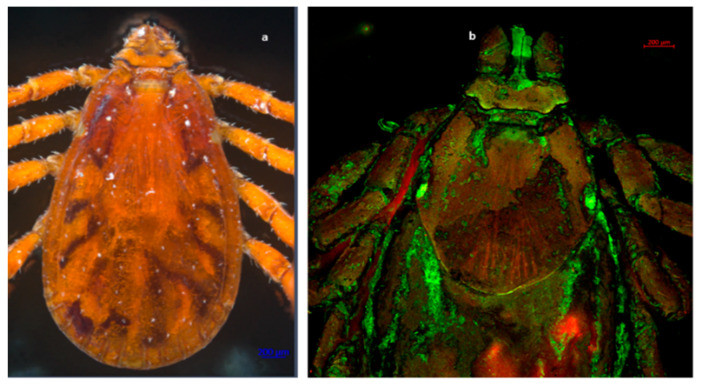
Dorsal view of a male tick of *R. sanguineus* s. l. Observe the basis of the hexagonal-shaped gnathostome, with a complete scutum and the ocelli featuring at each end of the scutum. (**a**) *R. sanguineus* s. l. male. Photography was carried out with an Axio Zoom V6 microscope. (**b**) *R. sanguineus* s. l. female. Photography obtained with an LSM 700 confocal scanning microscope.

**Figure 4 pathogens-11-00552-f004:**
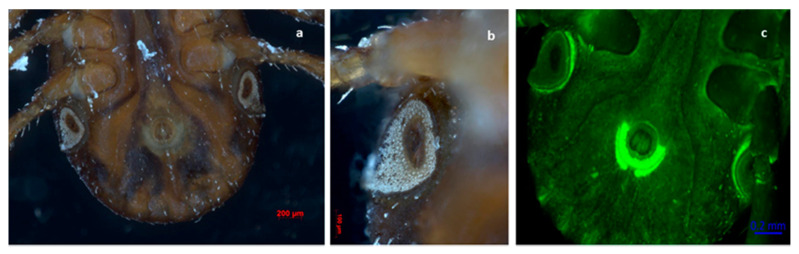
Rear ventral view of a semi-fully engorged female tick of *R. sanguineus* s. l. Observe the inverted U-shaped genital aperture and the spiracles below the fourth pair of legs. Festoons on the back side of the specimen can be seen. (**a**,**b**) Photography obtained with an Axio Zoom V6 microscope. (**c**) Photography obtained with a laser scanning confocal 700 (LSM 700).

**Figure 5 pathogens-11-00552-f005:**
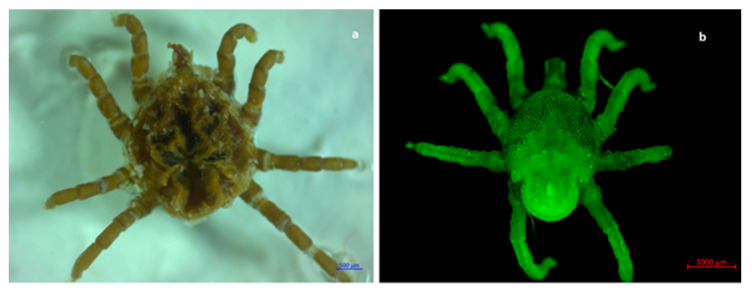
Dorsal view of *Otobius megnini* nymphs. Characterized by the spines contained throughout the body, specimen in (**b**) shows the spines-like processes are in the first half of the idiosoma. (**a**) Photography obtained with a Stemi 200C microscope. (**b**) Photography obtained with an Axio Zoom V6 microscope.

**Figure 6 pathogens-11-00552-f006:**
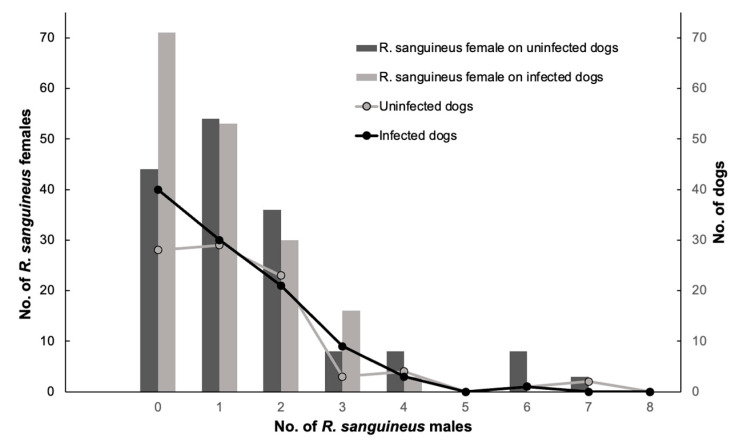
Abundance of *R. sanguineus* s. l. female ticks on dogs infected and uninfected for *E. canis* and its association to the dogs infested by a variable number of *R. sanguineus* s.l. males.

**Figure 7 pathogens-11-00552-f007:**
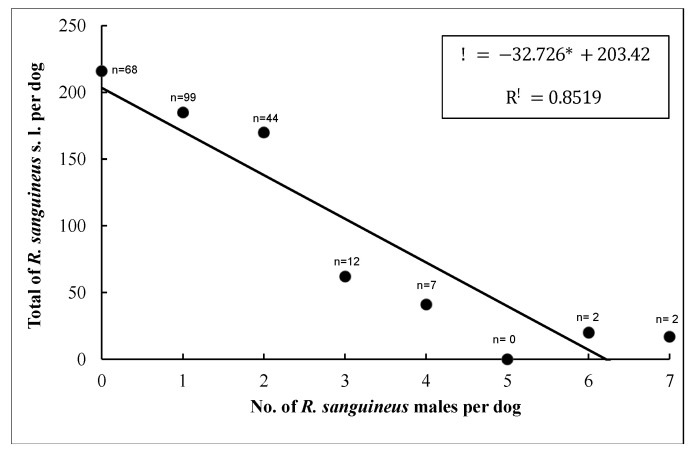
Relationship between the number of total ticks and number of male ticks per dog. Dots denote the number of dogs recorded for each class of number of male ticks, from zero up to seven. *n* = number of dogs.

**Figure 8 pathogens-11-00552-f008:**
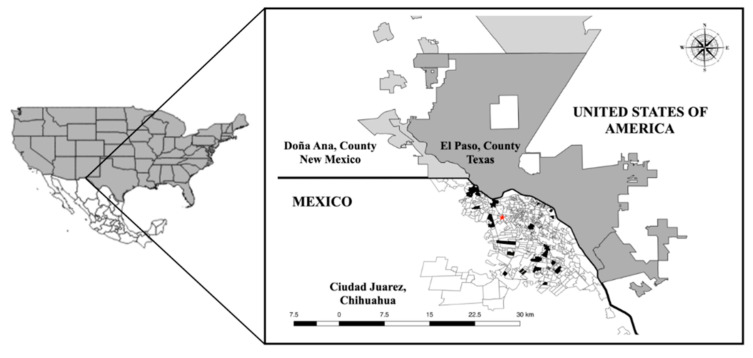
Geographic location of the binational transboundary metropolitan area between Mexico (in white with divisions: Juarez City, Chihuahua) and the United States of America (in dark gray: El Paso, County, Texas; in soft gray: Doña Ana, County, New Mexico). In Juarez City, the red star indicates the MARC, while the black areas show the locations of the neighborhoods where the VCs were located.

**Table 1 pathogens-11-00552-t001:** Incidence of *Ripicephalus*
*sanguineus* s. l. in 237 dogs examined in Juarez City, Chihuahua, Mexico in 2013–2014: Feeding status of females plus number of males and immature stages found in 137 FRD and 100 home dogs explored in both years.

Type and No. of Dogs	Year	Total	Females	Engorged Females	Unfed Females	Males	Nymphs	Larvae
FRD * 74	2013	402	150	105	45	186	56	0
63	2014	307	131	75	56	135	25	16
	Total	709	281	180	101	321	81	16
Home dogs 51	2013	147	56	27	29	81	19	0
49	2014	124	52	39	13	47	22	3
	Total	271	108	66	42	128	41	3
Total 237	Both years	980	389	246	143	449	122	19

* FRD: Free-roaming dogs.

**Table 2 pathogens-11-00552-t002:** Linear square mean (LSM) Standard error (SE) and Mean for each life metamorphic stage of ticks per year and both dog categories (FRD/home dogs), including the *p*-values from T-student and goodness of fit of chi-square/df.

Total Ticks					
Year	Mean	SE		Mean	SE
2013	3.9501	0.2381	FRD	5.1431	0.2713
2014	3.5180	0.2227	Home dogs	2.7020	0.2013
T value 1.34			Pr > |t| 0.1805		
Total Males	Mean	SE		Mean	SE
2013	1.9542	0.1731	FRD *	2.3048	0.1845
2014	1.4833	0.1492	Home dogs	1.2577	0.1398
T value 4.44			Pr < |t|0.0001		
Total Females	Mean	SE			
2013	1.4830	0.1119	FRD *	2.0517	0.1394
2014	1.4942	0.1182	House dogs	1.0801	0.1117
T value 5.19			Pr > |t| < 0.0001		
Engorged Females	Mean	SE		Mean	SE
2013	0.9339	0.09564	FRD *	1.3135	0.1098
2014	0.9283	0.09944	House dogs	0.6600	0.08617
T value 4.44			Pr > |t| <0.0001		
Empty Females	Mean	SE		Mean	SE
2013	0.5525	0.08021	FRD *	0.7373	0.09191
2014	0.5610	0.08594	House dogs	0.4203	0.08594
T value 2.59			Pr > |t| 0.0103		
Total Nymphs	Mean	SE		Mean	SE
2013	0.5650	0.1344	FRD *	0.5672	0.1268
2014	0.1247	0.1063	House dogs	0.4143	0.1127
T value 0.89			Pr > |t| 0.3760		

* FRD: Free-roaming dogs.

**Table 3 pathogens-11-00552-t003:** Sequences of primer sets and protocols used for PCR detection.

Pathogen	Oligonucleotide Sequence (5′ to 3′)	PCR Protocol	References
*E. canis*	ECC-AGAACGAACGCTGGCGGCAAGCC ECB- CGTATTACCGCGGCTGCTGGC	Initial denaturation at 94° for 1 min, followed by 35 cycles, each consisting of 94 °C for 1 min, 60 °C for 1 min, 72 °C for 40 s; and final extension at 72 °C for 3 min	[75]
*E. canis* Nested PCR	HE-TATAGGTACCGTCATTATCTTCCCTAT ECA-CAATTATTTATAGCCTCTGGCTATAGGAA	Initial denaturation at 94 °C for 1 min, followed by 35 cycles, each consisting of 94 °C for 1 min, 60 °C for 30 s, 72 °C for 40 s; and final extension at 72 °C for 3 min of final elongation	[75]
*A. platys*	8F-AGTTTGATCATGGCTCAG 1448R-CCATGGCGTGACGGGCAGTGT	Initial denaturation at 94 °C for 1 min, followed by 35 cycles, each consisting of 94 °C for 1 min, 45 °C for 1 min, 72 °C for 40 s; and final extension at 72 °C for 3 min	[76]
*A. platys* Nested PCR	PLATYSF-GATTTTTGTCGTAGCTTGCTATG EHR162R-TAGCACTCATCGTTTACAGC	Initial denaturation at 94 °C for1 min, followed by 35 cycles, each consisting of 94 °C for 1 min, 53 °C for 30 s, 72 °C for 40 s; and final extension at 72 °C for 3 min	[76]
*A.* *phagocytophilum*	GE3F-CACATGCAAGTCGAACGGATTATTC GE10R-TTCCGTTAAGAAGGATCTAATCTCC	Initial denaturation at 95 °C for 2 min, followed by 40 cycles, each consisting of 94 °C for 30 s, 55 °C for 30 s, 72 °C for 1 min; and final extension at 72 °C for 5 min	[77]
*A. phagocytophilum* Nested PCR	GE9F-AACGGATTATTCTTTATAGCTTGCT GE2R-GGCAGTATTAAAAGCAGCTCCAGG	Initial denaturation at 95 °C for 2 min, followed by 30 cycles, each consisting of 94 °C for 30 s, 55 °C for 30 s, 72 °C for 1 min; and final extension at 72 °C for 5 min	[77]
*R. rickettsii*	Rr190.70P-ATGGCGAATATTTCTCCAAAA Rr190.701N-GTTCCGTTAATGGCAGCATCT	Initial denaturation at 95 °C for 5 min, followed by 35 cycles, each consisting of 95 °C for 30 s, 58 °C for 30 s, 65 °C for 45 s; and final extension at 72 °C for 7 min	[78]
*R. rickettsii* Semi nested PCR	Rr190.70P-ATGGCGAATATTTCTCCAAAA Rr190.602N-AGTGCAGCATTCGCTCCCCCT	Initial denaturation at 96 °C for 30 s, followed by 35 cycles, each consisting of 94 °C for 30 s, 58 °C for 30 s, 72 °C for 45 s; and final extension at 72 °C for 7 min	[78]
*B. burgdorferi* s. l.	LY2F-GAAATGGCTAAAGTAAGCGGAATTGTAC LY2R-CAGAAATTCTGTAAACTAATCCCACC	Initial denaturation at 94 °C for 4 min, followed by 40 cycles, each consisting of 94 °C for 45 s, 55 °C for 45 s, 72 °C for 45 s; and final extension at 72 °C for 7 min	[79]

## Data Availability

Data are available from the authors upon reasonable request.

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
