# Peer review of "Possible Association between Selected Tick-Borne Pathogen Prevalence and Rhipicephalus sanguineus sensu lato Infestation in Dogs from Juarez City (Chihuahua), Northwest Mexico–US Border"

_pathogens, 2022, doi:10.3390/pathogens11050552_

Round 1
Reviewer 1 Report
This an interesting and well-written study investigating the prevalence of pathogens in ticks parasitising on domestic and stray dogs, as well as the prevalence of tick-borne pathogens in dogs.
Lines 111-116
The description of the GLIMMIX model fits better in the 2.5 section or the methods section.
Line 116
The mean number of ticks found on stray dogs was higher, compared to home dogs, according to Tables 1 and 2.
Table 2.
Stay or stray dogs?
Pathogen coinfection in tick pools described in line 143 and then stated in the conclusions section is a bit misleading. It is possible that within one tick pool there were two ticks infected with two different pathogens and this does not mean coinfection, but just codetection. This observation should be rephrased and made clear to avoid misinterpretation.
Lines 283-294
The explanation why male ticks were found to be associated with increased risk of E. canis infection is not clear. This paragraph ends with a sentence about female ticks and increased risk of E. canis transmission. Is that correct?
Lines 321-333
The procedure of dog recruitment is not clear. Could the authors make it clear how many dogs had their blood taken? In the 2.3 section one can read that 194 dogs were tested, whereas in 2015 there were 195 dogs. Importantly, more information about dogs recruited in 2015 should be provided. Were those dogs healthy? Why were they consulted in veterinary clinics? Were dogs with symptoms of tick-borne infection included or excluded from the analysis?
Also, it is not clear how many ticks were analyzed for the presence of pathogens. Ticks collected from 144 dogs were then pooled into 60 pools of 5 ticks each, indicating that there were probably 300 ticks, which gives a lower mean number of ticks per dog (2.08), compared to the mean number in table 2 (2.70). Is that correct? Could the authors describe how exactly ticks were pooled? Randomly, by sex, by metamorphic stage? If a non-random pooling was done, then were there any differences between each pool type?
Line 331
“Ticks from each dog were sorted as shown below” – but this was not shown.
Line 390
The sentence “Infestation of ticks…” looks unfinished.
Author Response
Dear editorial committee
Pathogens editorial office,
We appreciate so much the comments and suggestions. Thank you andthe anonymous reviewers for the comments and suggestions on our manuscript “Association between select tick-borne pathogen prevalence and Rhipicephalus sanguineus sensu lato infestation in dogs from Juarez City (Chihuahua), northwest Mexico-US border “, pathogens- 1665365.
The comments have been very helpful, and we made changes in response to the referee's suggestions. We now present a modified version of the manuscript hoping that you find it suitable for publication. The specific responses to each of the reviewer's are listed in attachment.

Reviewer 2 Report
Comments to authors: The MS presents interesting data on the circulation of pathogens associated with dog and dog ticks in an urban area of Mexico. Although the methods are suitable for the research objectives, several points must be reviewed so that the work has a real validity, according to the most current knowledge.
L29+. Rhipicephalus sanguineus is not a single species, it is a complex of species that originate in Europe (Rh. sanguineus s.s.) and Africa (Rh. sanguineus s.l., perhaps more than one species). Therefore, the authors must use Rh. sanguineus s.l. or Rh. sanguineus complex, not Rh. sanguineus. This aspect is important and basic, since recent studies show that there are two species of the Rh complex. sanguineus in Mexico, with sympatry in northern Mexico (including Chihuahua). From a public health point of view, both species of Rh. sanguineus have different roles in the transmission of each pathogen mentioned in MS.
L32-33. To verify this premise, the authors should review the paper of Braga-Ordoñez et al. 2016. Rev. Med.UADY for Yucatan.
L49. There are species of ticks that feed exclusively on amphibians and reptiles and occasionally on warm-blooded vertebrates, so it is not correct to refer to the parasitic behavior of ticks as "occasionally" when referring to amphibians and reptiles. Occasionally it is a term that could be more specifically used for an alternative parasitism in certain species.
L57-58. Suggestion, since there are several reports on these issues for Mexico and Latin America, I believe that the MS should be based on tick and/or dog monitoring studies in Mexico or Latin America. This is because the conditions are more similar and would be more representative, not only in terms of eco-epidemiology, but also in research conditions.
L60-62. Similar suggestion to the previous comment, there are many references to monitoring TBD in ticks from Mexico or Latin America, which could be more appropriate for the reasons described above.
L75-75. Italics for scientific names. In general, authors should review all scientific names.
Two points about IDs:
- It is important to state with certainty which species of the Rh sanguineus complex was or were found in this study since Chihuahua is the area of sympatry for the two species of the complex in Mexico. To which Genbank sequences were the amplified and sequenced products aligned?
- Where did they get Otobius? Ear canal? Of feral dogs or domestic dogs?
L122. No italics s.l.
T2. What are "stay dogs"?
L129-136. Did any of these dogs have symptoms for any of these diseases? Again, italics for scientific names.
L139-147. Italics for scientific names.
L147. How were the data described above obtained if the amplicons for Anaplasma phagocytophilum were not sequenced?
L171. As explained in previous lines, it must be made clear which species of the Rh complex. sanguineus was found in the study.
L181-182. From the morphological and molecular re-description of Rh. sanguineus s.s., and with the new molecular records of Rh. sanguineus s.l. from various countries, including Mexico, these new sequence references should be taken as the basis for separating Rh. sanguineus s.s. of Rh. sanguineus s.l.
L183-192. It is well known that the Rh complex. sanguineus mainly parasitizes dogs, so pointing out this fact is redundant. This paragraph may be reduced or deleted.
L193-198. The authors ignore the differences between Rh. sanguineus s.s. (European origin and reported in America in Mexico, USA and the Southern Cone of South America) with Rh. sanguineus s.l. (origin is probably Africa, it may represent more than one species and has been reported from Mexico to Brazil).
This text, and in general throughout the MS, should be modified and the recommendation for authors to review Nava et al. (2018). TTBDIS. https://doi.org/10.1016/j.ttbdis.2018.08.001 and Sánchez-Montes et al. (2019). TTBDIS. https://doi.org/10.1016/j.ttbdis.2019.04.013.
L203. O. megnini.
L226-227. Was it or was it not sequenced? How do you attribute the diagnosis?
L238. R. rickettsii.
L239-241. RMSF is an American disease, not a cosmopolitan one. The authors should compare their results (low infection rates) with the findings in other regions where R. rickettsii has been reported.
L243. In fact, it is the most important rickettsiosis in the entire continent.
L258-265. Correct the name B. burdorferi s.l. and s.l. without italics.
L283-234. This study was not experimental, so it cannot be confirmed that all stages were capable of transmitting E. canis.
Point 4.2. Could the authors give a further description of the study area, especially climate? Pointing out the number of the human´s population would also be important.
L326. Past clinical history or symptoms at the time of the study? Was the general condition of the dogs considered in the 2013-2014 samples, that is, in the animals without an owner?
L351-352. Why do the authors use a British key for a Mexican study? Although Rh. sanguineus s.l. is a complex of widely distributed species, regional keys are more appropriate, especially considering that at least two morphospecies (Rh. sanguineus s.s. and Rh. sanguineus s.l.) have been reported in northern Mexico.
Reference Nava et al. 2018; Sánchez-Montes et al. 2019.
Table 3. Review the scientific names, there are two misspelled.
L400. Again, it is important to determine which of the two Rh. sanguineus was present in the study and which was naturally infected by which pathogen.
Author Response
Dear editorial committee
Pathogens editorial office,
We appreciate so much the comments and suggestions. Thank you andthe anonymous reviewers for the comments and suggestions on our manuscript “Association between select tick-borne pathogen prevalence and Rhipicephalus sanguineus sensu lato infestation in dogs from Juarez City (Chihuahua), northwest Mexico-US border “, pathogens- 1665365.
The comments have been very helpful, and we made changes in response to the referee's suggestions. We now present a modified version of the manuscript hoping that you find it suitable for publication. The specific responses to each of the reviewer's are listed in attatchment.

Reviewer 3 Report
Title – rewrite to read as: Association between the prevalence of SELECTED tick-borne pathogens and Rhipicephalus sanguineus sensu lato infestation in dogs from Juarez City (Chihuahua), northwest Mexico-US border
Abstract – one decimal place seems to be enough for percentages (99.8%)
L30 – Cd. Juárez or Juarez city? Standardize
L34 – delete comma
L36-L37 – not clear why male ticks could increase the possibility of transmission
Keywords – display alphabetically
L43 – write disease names in lowercase
Remark 1: ticks do not transmit TBBD; ticks transmit the agents of those diseases – please adapt text throughout the manuscript
L62 – xenosurveillance
L75 – write out Rhipicephalus sanguineus [italic type] sensu lato at its first use in the main text; then abbreviate it as R. sanguineus [italic type] s.l.
L76 – Otobius megnini – italic type
Figure legends, etc. – always use italic type for Latin names
L87 – replace idesome with idiosoma
L109 – Which criteria have determined the numbers (137 + 100) of sampled dogs?
L112 – explain what GLIMMIX is
L116 – is this LSM the same as in L83? The meaning of LSM should be explained the first time it appears in the main text
Author Response
March 30, 2022.
Universidad Autónoma de Ciudad Juárez,
Dear editorial committee
Pathogens editorial office,
We appreciate so much the comments and suggestions. Thank you andthe anonymous reviewers for the comments and suggestions on our manuscript “Association between select tick-borne pathogen prevalence and Rhipicephalus sanguineus sensu lato infestation in dogs from Juarez City (Chihuahua), northwest Mexico-US border “, pathogens- 1665365.
The comments have been very helpful, and we made changes in response to the referee's suggestions. We now present a modified version of the manuscript hoping that you find it suitable for publication. The specific responses to each of the reviewer's are listed in attachment.

Reviewer 4 Report
The present manuscript is devoted to the study of tick-borne bacteria prevalence in dogs and infesting ticks in Juarez, Chihuahua, on the Mexico-U.S. border. The ethical policies were followed. Authors also made attempts to identify the factors that influence on the E. canis prevalence in dogs. I agree with authors that it’s important to carry out epidemiological studies in dogs due to epidemiological significance of ticks, in particular Rhipicephalus sanguineus. The results and discussion of dogs’ examination on the presence of tick-borne bacteria is very valuable, and I strongly advise to shift the focus of the manuscript to this part of the work.
The language of the manuscript should be checked, as there are some logic, grammar and spelling mistakes.
General comments:
You cannot detect a disease in a tick or a host, only disease agent. There is lots of “TBBDs detection” in the text, please correct this on TBB detection or something else.
Check the spelling and italicize the species and genus names of ticks and tick-borne bacteria (do not italicize sensu lato)
“infestation of R. sanguineus” (quite often in text) – I think it’s wrong (I’m not an expert in English). Infestation of animals with (or by) ticks
Section 2.1.: The photos of ticks are beautiful, but I do not see the need to show the morphological characteristics of ticks in such detail since you used DNA barcoding.
Section 2.2: why don't you analyze the distribution of ticks among dogs here (tick burden) (ideally by developmental phases)? It would be very informative if you overlaid these histograms with negative binomial distribution and demonstrated the correlation. It would be even more interesting if you analyse the tick burden using your GLIMMIX model with dogs’ breed, sex and age as predictors. After all, there is information about the relationship of sex and breed of dogs with tick infestation (Dantas-Torres, 2010), and you could confirm or refute these observations.
Table 2 - it is not clear what the T value refers to. From the materials and methods, I realized that this is most likely a comparison of groups FRD and home dogs, but this should be clear from the table. It is also not clear what Pr and |t| refer to.
All variables and abbreviations should be described in the footnotes to the table (LSM, SE, Mean, etc).
In the text you accepted the names of dog groups as FRD and home dogs, and in the table you have stray and home dogs.
You are mentioning in the text larvae, but I don’t see them in the table.
I noticed an interesting difference in nymph infestation of dogs in 2013 and 2014. Maybe it should be noted in the text?
section 2.4: you can calculate the actual tick infection rate, not just the percentage of positive pools. To do this, you can use for example this online resource: https://epitools.ausvet.com.au/pooledprevalence.
Did you analyze only adult ticks or all phases of development? It is written in the materials and methods that you formed pools taking into account the phase of development and sex of ticks. If so, were there any differences in the infection rate of males, females, and immature ticks?
You cannot write about co-infections in ticks, because you analysed ticks in pools! You need to study individuals to talk about co-infections (as you did with dogs).
section 2.5: a lot of questions and comments to this part.
- you should mention the risk factors that you are studying. Searching the information in materials and methods can be quite irritating.
- You used many interesting predictors in your model. It will be very useful to see all results of your regression analysis in the supplementary.
- as I understood, the only predictor that somehow affected the infection rate of dogs with E. canis was the number of male ticks. However, the significance level was only 0.09. With all my respect to the obtained results, with this significance you can only carefully note the probable association of the male ticks number with the E. canis prevalence in dogs, while in the manuscript you confidently assert about the observed association. You even put it in the title of the manuscript even though most of the discussion in the article is devoted to tick-borne bacteria detection in dogs. I suggest changing the title, or at least adding the word “possible”.
- I don’t understand figure 6 at all. What do the bars show – the mean number of females per dog? If so, then where are the confidence intervals? Or it’s a total amount of females on dogs with corresponding number of males? And what does the positive/negative dog graph show? Is this the number of dogs? Then a second axis is needed, displaying the number of dogs. Unfortunately, in its current form, figure 6 does not show any association of dog infection rate with the number of male ticks.
- figure 7 – you need the second axis for your bars, indicating the number of dogs. And what are the black cyrcles?
Discussion:
line 283: “…all parasitic stages of R. sanguineus were capable of transmitting E. canis in dogs.” – this is a very bold statement! Which of your results allowed you to conclude this?
lines 284-300: I don’t see your point here… the title of the manuscript is about association between infection rate and infestation. Most of the discussion is devoted to the comparison of the tick-borne bacteria prevalence in dogs with that in the nearby territories. And in the end it’s stated that males can influence the E. canis prevalence in dogs because of their capability in transstadial transmission and that males can transmit this bacteria without females. But females (and maybe even immature stages) can do all this too. And also females feed longer plus the last paragraph in the discussion equals that females are better vectors of E. canis than males… What should be mentioned here is that unlike females or immature ticks, male ticks can take multiple blood meals, remain for long periods of time on the host and increase the feeding performance of nymphs (Dantas-Torres, 2010).
Specific comments:
line 75: you need to write full species name here, as it is the first time you mention it in the main text. Also check here and throughout the text (especially in figures names) that the species and genus names are italicized.
line 116: “…was significantly lower than…”, I think you meant “higher” here
line 119: ± is missing
lines 329-330: these 144 dogs are home dogs from the study of the year 2015? it’s not clear from the text
line 331: how the ticks were sorted? I couldn’t find anything below
line 332: were collected
To sum up, this manuscript has valuable information about tick-borne bacteria prevalence in dogs and ticks.
Author Response
Dear editorial committee
Pathogens editorial office,
We appreciate so much the comments and suggestions. Thank you andthe anonymous reviewers for the comments and suggestions on our manuscript “Association between select tick-borne pathogen prevalence and Rhipicephalus sanguineus sensu lato infestation in dogs from Juarez City (Chihuahua), northwest Mexico-US border “, pathogens- 1665365.
The comments have been very helpful, and we made changes in response to the referee's suggestions. We now present a modified version of the manuscript hoping that you find it suitable for publication. The specific responses to each of the reviewer's are attachment.

Round 2
Reviewer 2 Report
The new version includes several improvements in the text; however, it is not clear how to differentiate between the two species of the R. sanguineus complex that co-inhabit northern Mexico. This impressision could mean that pools of both species were blended, instead of separating them. Why don't the authors show the results of the molecular identification analyzes they carried out (L77-79)?
The molecular and morphological identification of the sanguineus complex is available since Nava´s et al. (2018) re-description, and it was possible to separate them in the work by Sánchez-Montes et al. (2019).
Another aspect to consider in point 4.2.2 is that no symptoms are linked to any of the animals, which leaves doubt as to whether they were infected, since the prevalence of microorganisms cannot be associated with the mere presence of ticks.
The results are not correctly approached, there are no sequences to compare, there are no reference sequences (except the accession numbers of the results in ticks). This leaves doubts about the results and their interpretation, especially when no symptoms were recorded in the dogs.
Grammar should be checked (e.g. L303, 333). Lines 112-115 have the explanations of the acronyms in non-corresponding paragraphs.
Author Response
April 25, 2022.
Dear editorial committee
Pathogens editorial office,
We appreciate so much the anonymous reviewers comments and suggestions on our manuscript “Association between select tick-borne pathogen prevalence and Rhipicephalus sanguineus sensu lato infestation in dogs from Juarez City (Chihuahua), northwest Mexico-US border “, pathogens- 1665365.
The comments have been very helpful, and we made changes in response to the referee's suggestions. We now present a modified version of the manuscript hoping that you find it suitable for publication. The specific responses to each of the reviewer's can be found in an attachment.

Reviewer 4 Report
The authors of this article tried to take into account all my comments and the text has become much better. However, I still have some questions and corrections.
line 26 - pathogens (TBBPs)
line 104 - space missing
line 115 - delete space after LSM
Table 1 and 2 - can you format the tables, please, so they are neater and easier to read? For example, in Table 1, justify the text on the lines so that the number of dogs is on the same level as the rest of the text. In table 2, make sure that the T value is on the left and Pr is in the middle.
section 2.4: here I suggested you to calculate pathogen prevalence - the percent of positive ticks (not pools). But since you are not discussing this result in the manuscript, you can ignore my comment.
section 2.5: sorry, but your supplementary is unreadable. I wanted to see the results of the regression analysis, not the SAS output. You need to translate it in English and present it in understandable form (please, see any article on similar topic). I haven't found the significance of the total amount of ticks per dog on E.canis prevalence in the supplementary. Please, respect the reader and present your data in a way that makes it easy to figure out which factor is which and it's significance.
line 151 - could influence (alpha 0.09 is not very significant in my opinion)
Fig. 6. - All I see on this figure is that the presence of males on dogs somehow decreases the infection rate with E. canis. The second axis showing the number of dogs is needed. We need to know the number of dogs in each point.
Fig. 7 - when you changed the influence of males to females, you changed your main idea. And this figure doesn't correspond with this new idea. Rethink your results and make adequate figures.
line 165 - I think you meant black cyrcles here instead of "unfilled bars"
line 166 - n - number of dogs
line 200, 212 - TBBP
lines 279-296 - that doesn't make sense. You corrected that females increase the possibility of E. canis transmission, which is understandable - they feed for a long time, and bacterial agents need time to be transmitted during bloodfeeding. Why are you discussing males? Again, you changed your main idea, so change the discussion!
lines 319-320 - bad English, revise
lines 324-326 - bad English
line 333 - space needed
lines 361-365 - this text belongs to the Statistical analysis section
line 372 - maybe 'homogenized' is a better word here than 'macerated' (only suggestion)
line 424 - males or females?
To sum up, the authors in their response cited an misinterpretation of their results and now, according to their data, the abundance of females, not males, affects the E. canis infection in dogs. A change in the main idea implies a change in the discussion and understanding of the material, which, unfortunately, I did not see. The careful revision of the Section 2.5, the end of Discussion and readable Supplementary with results of the regression analysis are needed.
Author Response
Dear editorial committee
Pathogens editorial office,
We appreciate so much the anonymous reviewers comments and suggestions on our manuscript “Association between select tick-borne pathogen prevalence and Rhipicephalus sanguineus sensu lato infestation in dogs from Juarez City (Chihuahua), northwest Mexico-US border “, pathogens- 1665365.
The comments have been very helpful, and we made changes in response to the referee's suggestions. We now present a modified version of the manuscript hoping that you find it suitable for publication. The specific responses to each of the reviewer's are in an attachment.
Kind regards.
